# Survey on the association between *Toxoplasma gondii* infection and violent behavior in inmates

Adriana Rocha-Salais[1,2], Fátima Yazmin Muñoz-Larreta[3], Sergio Ignacio García-Pérez[4], Alejandro Israel Serrato-Enríquez[1], Manuel Arturo Rivas-González[5], Antonio Sifuentes-Alvarez[1], Elizabeth Rábago-Sánchez[1], Isabel Beristain-García[6], Cosme Alvarado-Esquivel[1]*

1 Faculty of Medicine and Nutrition, Biomedical Research Laboratory, Juárez University of Durango State, Durango, Mexico, 2 Instituto Mexicano del Seguro Social, Unidad de Medicina Familiar No. 44, Durango, Mexico, 3 Centro de Reinserción Social No. 1 de Durango, Durango, Mexico, 4 Hospital Nueva Vizcaya, Centro Quirúrgico Endoscópico, Durango, Mexico, 5 Instituto Mexicano del Seguro Social, Unidad de Medicina Familiar No. 49, Durango, Mexico, 6 Facultad de Enfermería y Obstetricia, Juárez University of Durango State, Durango, Mexico

* alvaradocosme@yahoo.com

**Data Availability Statement:** All relevant data are within the manuscript and its Supporting information files.

## Abstract

We evaluated the association between *T. gondii* seropositivity and violent behavior in a sample of inmates in Durango, Mexico. Through a cross-sectional study design, we studied 128 inmates (mean age: 35.89 ± 10.51; range: 19–65 years). Sera of participants were analyzed for anti-*T. gondii* IgG antibodies using a commercially available enzyme-linked immunosorbent assay. Violence was assessed by 1) the Historical, Clinical and Risk Management-20 (HCR-20) tool; 2) the type of the crime for which inmates were convicted; and 3) the Buss-Perry Aggression Questionnaire (AGQ). Of the 128 inmates, 17 (13.3%) had high risk of violence by the HCR-20 criteria, 72 (56.3%) were considered violent by the type of the crime committed, and 59 (46.1%) were considered violent by the AGQ. Depending on the evaluation method of violence, the seroprevalence of *T. gondii* infection in violent inmates varied from 0% to 6.9%. No statistically significant difference in anti-*T. gondii* IgG seroprevalence between violent and non-violent inmates was found (for instance by AGQ, OR: 1.17; 95% CI: 0.22–6.07; *P* = 1.00). Mean scores of the AGQ in *T. gondii* seropositive inmates (73.67 ± 29.09; 95% CI: 50.00–99.31) were similar to those (79.84 ± 25.00; 95% CI: 75.46–84.27) found in *T. gondii* seronegative inmates (*P* = 0.55). Mean scores of anger, psychical aggression, verbal aggression, and hostility in *T. gondii* seropositive inmates were similar to those found in *T. gondii* seronegative inmates. Results of this study suggest that infection with *T. gondii* is not associated with violence in inmates in Durango, Mexico. Further studies with larger sample sizes and in several correctional facilities to determine the association between *T. gondii* infection and violence in inmates are needed.

**Funding:** This study was financially supported by Juarez University of Durango State, Mexico. The funders had no role in the study design, data collection and analysis, decision to publish, or preparation of the manuscript.

**Competing interests:** The authors have declared that no competing interests exist.

**Abbreviations:** AGQ, Buss-Perry aggression questionnaire; CI, Confidence interval; FAF, Questionnaire for Measuring Factors of Aggression; HCR-20, Historical, Clinical and Risk Management-20; OR, Odds ratio; SD, Standard deviation.

## Introduction

*Toxoplasma gondii* (*T. gondii*) is a protozoan parasite that causes infections in humans and animals [1]. Infections with *T. gondii* occur worldwide [2] and nearly one-third of humanity has been exposed to this parasite [3]. Transmission routes of *T. gondii* infection include ingestion of cysts in meat and meat derivates, milk contaminated with tachyzoites, oocysts in water, sand, and soil, and oocysts in raw fruits and vegetables [4]. Alternative routes for transmission are organ transplantation [5,6] and blood transfusion [5,7]. *T. gondii* can produce outbreaks [4]. Primary infection is usually subclinical but, in some patients, cervical lymphadenopathy or ocular disease can be present [8]. However, primary infections during pregnancies can cause blindness and mental retardation in congenitally infected children [3]. In addition, *T. gondii* infection can cause devasting disease in immunocompromised individuals [3]. *T. gondii* resides, in a latent form, in the central nervous system and drastically alters the behavior of rodents and is associated with the incidence of specific neuropsychiatric conditions in humans [9]. Infections with *T. gondii* have been associated with mental disorders as obsessive-compulsive disorder [10], schizophrenia [11,12], generalized anxiety disorder [13], mixed anxiety and depressive disorder [14], bipolar disorder [15], and depression [13,15]. Furthermore, high *T. gondii* antibody titer has been associated with suicide attempts [16,17].

There is scanty information about the association between of *T. gondii* infection and violent behavior. Infection with *T. gondii* was associated with nonfatal suicidal self-directed violence [18,19]. Aggression and impulsivity were associated with latent *T. gondii* infection in psychiatrically healthy subjects [20]. In another study, researchers found that *T. gondii* seropositivity and high blood kynurenine have a cumulative effect on the risk of nonfatal suicidal self-directed violence in schizophrenic patients [21]. In the present study, we investigated the question as to whether *T. gondii* infection might be associated with violent behavior in inmates. Therefore, we sought to evaluate the association between *T. gondii* seropositivity and violent behavior in a sample of inmates in the northern Mexican city of Durango.

## Materials and methods

### Study design and study population

A cross-sectional study of inmates at a state correctional facility in Durango, Mexico was performed. As a strategy to enroll inmates in the study, four open meetings with inmates of various sections of the correctional facility were organized. In these meetings inmates were informed about the study and invited to participate. Inclusion criteria for enrollment were: 1) inmates of the state correctional facility in Durango City; 2) aged 18 years and older; and 3) with a written informed consent to participate in the survey. In total, 236 inmates attended the meetings, 108 (45.8%) of them declined the invitation, and 128 (54.2%) were willing to participate in the survey. Participants included 16 females and 112 males. Mean age of participants was 35.89 ± 10.51 (range: 19–65) years.

### Violent behavior

Three different criteria for defining a violent behavior in inmates were used: 1) an evaluation based on the Historical, Clinical and Risk Management-20 (HCR-20) tool that assesses the risk of violence [22]; 2) an evaluation based on the type of the crime for which inmates were convicted; and 3) Assessment of aggression by the Buss-Perry aggression questionnaire (AGQ) [23–26]. HCR-20 assessments were performed at admittance of inmates to jail, and by using this tool the correctional facility classified inmates into three groups of violence risk: low (scores 0–11), medium (scores 12–21), and high (scores 22–30). Individual HCR-20 scores

were not provided by the correctional facility but provided a two-group classification: violent (high HCR-20 scores) and non-violent (HCR-20 scores lower than 22). Analysis was thus performed considering these two groups. Concerning the evaluation based on the types of crimes, violent inmates included participants convicted for aggravated assault, murder, rape, rape attempt, domestic and non-domestic violence, possession of weapons, wounding, kidnapping, and indecent behavior with people. Whereas inmates were considered as non-violent when they were convicted for burglary/robbery/theft, sale and possession of drugs, entering in a property without permission, corruption of minors, and fraud. The AGQ includes items to assess four dimensions of aggression: physical aggression, verbal aggression, anger, and hostility [23]. An inmate was considered as violent when he/she obtained an AGQ overall score of 83 or more. Whereas inmates who scored 82 and lower were considered as non-violent. The use of these questionnaires was a minimal cost methodology. General sociodemographic characteristics of the violent and non-violent inmates is shown in Table 1.

### Detection of anti-*T. gondii* IgG antibodies

Serum samples from participants were obtained and frozen down at -20° C until analyzed. A commercially available enzyme immunoassay kit: "*Toxoplasma* IgG" (Diagnostic Automation/Cortez Diagnostics Inc., Woodland Hills, CA, USA) was used for detection of anti-*T. gondii* IgG antibodies. The test was performed according to the instructions of the manufacturer. Negative and positive controls provided in the kit were included in each run.

### Statistical analysis

Statistical analysis was performed with the software SPSS version 20. Sample size was calculated with following values: a population size of 2000, an expected frequency of exposure of 6.1% [27], confidence limits of 5%, a design effect of 1.0, and 1 cluster. Thus, a sample size of 84 people was obtained. Comparison of sociodemographic data was performed by Pearson's

**Table 1. Sociodemographic characteristics of the study population.**

| Characteristic | 59 Violent inmates* | | 69 Non-violent inmates* | | *P* |
|---|---|---|---|---|---|
| | No. | % | No. | % | value** |
| Age groups (years) | | | | | |
| 30 or less | 24 | 40.7 | 25 | 36.2 | 0.49 |
| 31–50 | 31 | 52.5 | 35 | 50.7 | |
| >50 | 4 | 6.8 | 9 | 13 | |
| Gender | | | | | |
| Male | 51 | 86.4 | 61 | 88.4 | 0.73 |
| Female | 8 | 13.6 | 8 | 11.6 | |
| Birthplace | | | | | |
| Durango State | 47 | 79.7 | 54 | 78.3 | 0.84 |
| Other Mexican State | 12 | 20.3 | 15 | 21.7 | |
| Education | | | | | |
| No education | 0 | 0.0 | 1 | 1.4 | 0.61 |
| 1 to 6 years | 6 | 10.1 | 6 | 8.7 | |
| 7–12 years | 49 | 83.1 | 54 | 78.3 | |
| >12 years | 4 | 6.8 | 8 | 11.6 | |

*Determined by AGQ.

**Determined by Pearson's' Chi squared test.

chi squared test. The frequencies of *T. gondii* seropositive rates among the groups were compared with the two-sided Fisher's exact test, and the odds ratios (OR) and 95% confidence intervals (CI) were calculated. The AGQ scores among different groups were compared with the student´s *t* test. Logistic regression analysis with the Enter method was used to further determine the association between *T. gondii* infection and violence and sociodemographic characteristics of inmates. Age- and gender-adjusted OR and 95% CI were calculated. A *P* value < 0.05 was considered as statistically significant. Normality distribution of AGQ scores was measured by the Shapiro-Wilk test. The result (sig) of this test was 0.02, indicating that the data deviate from a normal distribution.

## Ethical aspects

This study was approved by the Ethics Committee of the Faculty of Medicine and Nutrition of Juárez University of Durango State, Mexico (Approval No: CEI-A-2018-01). All participants were informed about the aims and procedures of the survey. Participation was voluntary, and a written informed consent from all participants was obtained. Participants were able to opt out of the study. Inmates who did not participate received the same treatment offered to participants.

## Results

Anti-*T. gondii* IgG antibodies were found in 6 (4.7%) of the 128 inmates studied. With respect to the HCR-20 criteria, 17 (13.3%) of the 128 inmates had high risk of violence. Whereas using the evaluation based on the type of the crime for which inmates were convicted, 72 (56.3%) of the 128 inmates were considered violent. Concerning the AGQ, 59 (46.1%) of the 128 inmates were considered violent. Results of the rates of *T. gondii* seropositivity and violence in inmates are shown in Table 2. Of the 128 inmates, 113 (88.3%) were considered as violent in at least one of the evaluations, 50 (39.1%) in at least 2 evaluations, and 4 (3.1%) in the 3 evaluations. Depending on the evaluation method to detect violence, the seroprevalence of *T. gondii* infection in violent inmates varied from 0% to 6.9%. No statistically significant difference in anti-*T. gondii* IgG seroprevalence between violent and non-violent inmates was found. This was the case when inmates were considered violent in at least one evaluation (HCR-20, crimes, or AGQ), in at least two evaluations or in the three evaluations.

**Table 2. Association between *T. gondii* seropositivity rates and violence groups.**

| Evaluation | Violent inmates | | | Non-violent inmates | | | | | |
| --- | --- | --- | --- | --- | --- | --- | --- | --- | --- |
| | | Seroprevalence of | | | Seroprevalence of | | | | |
| | No. | *T. gondii* infection | | No. | *T. gondii* infection | | Odds | 95% confidence | *P* |
| | tested | No. | % | tested | No. | % | ratio** | interval | value[¶] |
| HCR-20 | 17 | 0 | 0 | 111 | 6 | 5.4 | 0.46 | 0.00–5.73[¶] | 1.00 |
| Crimes | 72 | 5 | 6.9 | 56 | 1 | 1.8 | 4.10 | 0.46–36.18 | 0.22 |
| AGQ | 59 | 3 | 5.1 | 69 | 3 | 4.3 | 1.17 | 0.22–6.07 | 1.00 |
| No. of positive evaluations* | | | | | | | | | |
| In at least 1 | 113 | 6 | 5.3 | 15 | 0 | 0.0 | 1.87 | 0.00–6.68[¶] | 1.00 |
| In at least 2 | 50 | 2 | 4 | 78 | 4 | 5.1 | 0.77 | 0.13–4.37 | 1.00 |
| In all 3 | 4 | 0 | 0 | 124 | 6 | 4.8 | 2.02 | 0.00–35.82[¶] | 1.00 |

*Detection of violence by HCR-20, Crimes, or AGQ.

[¶]Determined by Fisher's exact test.

**Table 3. Association between *T. gondii* seropositivity rates and AGQ scores.**

| Type of aggression | Scores in inmates with | | | | Scores in inmates with | | | | *P* value* |
| | *T. gondii* positive test | | | | *T. gondii* negative test | | | | |
| | No. tested | Mean | SD | Range | No. tested | Mean | SD | Range | |
|---|---|---|---|---|---|---|---|---|---|
| Anger | 6 | 15.33 | 5.57 | 9–23 | 122 | 19.48 | 7.12 | 7–35 | 0.16 |
| Physical aggression | 6 | 25.00 | 9.94 | 13–37 | 122 | 25.4 | 9.34 | 9–43 | 0.91 |
| Verbal aggression | 6 | 11.50 | 6.62 | 5–21 | 122 | 12.55 | 5.42 | 5–25 | 0.64 |
| Hostility | 6 | 21.83 | 9.96 | 12–40 | 122 | 22.41 | 9.39 | 8–40 | 0.88 |
| All | 6 | 73.67 | 29.09 | 48–119 | 122 | 79.84 | 25.00 | 31–134 | 0.55 |

*Determined by Fisher's exact test.

Table 3 shows the results of the AGQ scores found in inmates and the seroprevalence of *T. gondii* infection. Mean scores of the AGQ in *T. gondii* seropositive inmates (73.67 ± 29.09; 95% CI: 50.00–99.31) were similar to those (79.84 ± 25.00; 95% CI: 75.46–84.27) found in *T. gondii* seronegative inmates (*P* = 0.55). Stratification by the type of aggression showed that the mean scores of anger, psychical aggression, verbal aggression, and hostility in *T. gondii* seropositive inmates were similar to those found in *T. gondii* seronegative inmates.

Violence and sociodemographic characteristics of inmates did not associate with *T. gondii* infection by logistic regression analysis (Table 4).

## Discussion

Little is known about the link between *T. gondii* infection and violence. Infection with *T. gondii* has been found associated with self-directed violence in people in Sweden [19], mothers in Denmark [18], and schizophrenic patients in the USA [21]. In addition, infection with *T. gondii* was associated with aggression and impulsivity in psychiatrically healthy adults in Germany [20]. However, the association between infection with *T. gondii* and violence has not been studied in inmates. Inmates may be considered as a population group with high rates of violence. Many inmates have been convicted for violent crimes. Therefore, in the present study we sought to determine the association between *T. gondii* seropositivity and violent behavior in a sample of inmates in a state correctional facility in Durango, Mexico. We found that the seroprevalence of *T. gondii* infection did not vary among violent and non-violent inmates as detected by the HCR-20, crimes committed, or AGQ. Results thus suggest that infection with *T. gondii* is not associated with violence in inmates in our setting. Apparently, our results conflict with those found in other studies on the positive association between *T. gondii* infection and violence. However, comparison of our results with those found in other studies should be

**Table 4. Logistic regression analysis of characteristics of inmates and their association with *T. gondii* infection.**

| Characteristic | Odds ratio** | 95% confidence interval | *P* value |
|---|---|---|---|
| Violence* | 0.57 | 0.09–3.50 | 0.54 |
| Birthplace | 0.23 | 0.04–1.42 | 0.11 |
| Education | 0.27 | 0.60–1.28 | 0.10 |

*Determined by AGQ.

**Adjusted by age and gender.

interpreted with care since difference on the characteristics of the study populations, types of violence, and criteria for the evaluation of violence among the studies exists. Firstly, the positive association between *T. gondii* infection and self-directed violence found in people in Sweden [19], mothers in Denmark [18], and schizophrenic patients in the USA [21] cannot be fairly compared with the lack of association found in our study. This is because we studied violence of inmates against other people whereas other researchers studied self-directed violence [18,19,21]. In addition, our study population has different characteristics as gender and clinical diagnosis from the population groups of other studies including mothers [18], schizophrenic patients [21], and people with and without suicide behavior [19]. On the other hand, in a study of psychiatrically healthy adults, researchers evaluated aggression with the Questionnaire for Measuring Factors of Aggression (FAF) [20] whereas we evaluated aggression with the AGQ.

We found a low seroprevalence of *T. gondii* infection in inmates in general, and this finding was unexpected since a high (21.1%) seroprevalence of *T. gondii* infection in inmates was previously reported [28]. In addition, we found a low seroprevalence of *T. gondii* infection in violent inmates. This low seroprevalence in violent inmates is comparable with a 6.1% seroprevalence of *T. gondii* infection reported in the general population in Durango City, Mexico [27]. In addition, the low seroprevalence found in violent inmates in this study is lower than those reported in other populations groups in the same Durango City including waste pickers (21.1%) [29], schizophrenic patients (20%) [12], and female sex workers (15.4%) [30]. The lack of association between IgG seropositivity to *T. gondii* and violence found in this study does not necessarily mean that *T. gondii* infection might not be linked to violence. We assessed violence by using three methods and determined *T. gondii* infection by IgG serology but more methods to assess violence and to determine *T. gondii* infection including molecular methods can be used to determine the association between *T. gondii* infection and violence in inmates. Further studies to confirm or challenge our results are needed.

This study has some limitations including a small sample size, a low prevalence of *T. gondii* infection among the studied population and was performed in only one correctional facility. Further studies with larger sample sizes and in several correctional facilities to determine the association between *T. gondii* infection and violence in inmates should be conducted.

## Conclusions

Results of this study using three types of evaluations of violence suggest that seropositivity to *T. gondii* is not associated with violence in inmates in Durango, Mexico. However, further studies with larger sample sizes and in several correctional facilities to determine the association between *T. gondii* infection and violence in inmates are needed.

## Supporting information

**S1 File. Date set of the study.**
(XLSX)

## Author Contributions

**Conceptualization:** Cosme Alvarado-Esquivel.

**Data curation:** Adriana Rocha-Salais, Fátima Yazmin Muñoz-Larreta, Sergio Ignacio García-Pérez, Alejandro Israel Serrato-Enríquez, Manuel Arturo Rivas-González.

**Formal analysis:** Antonio Sifuentes-Alvarez, Cosme Alvarado-Esquivel.

**Funding acquisition:** Cosme Alvarado-Esquivel.

**Investigation:** Adriana Rocha-Salais, Fátima Yazmin Muñoz-Larreta, Sergio Ignacio García-Pérez, Alejandro Israel Serrato-Enríquez, Manuel Arturo Rivas-González, Elizabeth Rábago-Sánchez, Isabel Beristain-García, Cosme Alvarado-Esquivel.

**Methodology:** Antonio Sifuentes-Alvarez, Elizabeth Rábago-Sánchez, Isabel Beristain-García, Cosme Alvarado-Esquivel.

**Writing – original draft:** Cosme Alvarado-Esquivel.

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
