## [Decision Letter · Decision Letter 0]

27 Dec 2022

PONE-D-22-31301Survey on the association between Toxoplasma gondii infection and violent behavior in inmatesPLOS ONE

Dear Dr. Alvarado-Esquivel,

Thank you for submitting your manuscript to PLOS ONE. After careful consideration, we feel that it has merit but does not fully meet PLOS ONE’s publication criteria as it currently stands. Therefore, we invite you to submit a revised version of the manuscript that addresses the points raised during the review process.

We look forward to receiving your revised manuscript.

Kind regards,

Masoud Foroutan, Ph.D; Assistant Professor

Academic Editor

PLOS ONE

Journal Requirements:

2. Please provide additional information regarding the considerations made for the prisoners included in this study. For instance, please discuss whether participants were able to opt out of the study and whether individuals who did not participate receive the same treatment offered to participants.

Reviewers' comments:

Reviewer's Responses to Questions

**Comments to the Author**

1. Is the manuscript technically sound, and do the data support the conclusions?

Reviewer #1: Partly

Reviewer #2: Yes

2. Has the statistical analysis been performed appropriately and rigorously? 

Reviewer #1: Yes

Reviewer #2: Yes

3. Have the authors made all data underlying the findings in their manuscript fully available?

Reviewer #1: Yes

Reviewer #2: Yes

4. Is the manuscript presented in an intelligible fashion and written in standard English?

Reviewer #1: Yes

Reviewer #2: Yes

5. Review Comments to the Author

Reviewer #1: Comments to Author:

This paper deals with to determine association between T. gondii seropositivity and violent behavior in 128 inmates in Durango, Mexico. Results in this study show that infection with T. gondii is not associated with violence in inmates in Durango, Mexico. It is a quite interesting research field, but observations (factors) and methods related to assessing effects are not enough suitable. To the reviewer's point of view, it is a well-written, well-organized, and well-structured paper. Hence, it can be published and for improve your manuscript I suggested below comments:

Abstract:

• Start the results with some descriptive statistics (mean ± SD, N (%), etc.) for participate (128 inmates).

• Please report more statistical results (coefficients, 95% CI’s, P-value, etc.) in abstract. Statistical results of associations.

• The actual P-value* should be expressed (P=.04) rather than expressing a statement of inequality (P<.05), unless P<.001.

• Methods: Are you measured the costs? Mention to it.

• What mean the “robust association”?

Materials & Methods:

• All acronyms/abbreviations must be explained in parenthesis after their first occurrence within each standalone section of a paper (abstract, main text, acknowledgements, figures, and tables). If any acronyms/abbreviations are used in the main text (excluding tables and figures), please compile them in an "Abbreviations" section at the end of the paper.

• Are you test the normality distribution for your quantities responses (HCR-20, Crimes, and AGO score). How you test the normality of data?

• Why did you not use multiple logistic regression analysis to examine the adjusted effects of confounding variables such as age and gender? Reported coefficients are crude effects and the effect of confounders in your study (statistical analyses) is ignored, which causes results with error [1].

Results:

• Report means of quantities variables with standard deviations in the text.

“Mean ± SD”

• For univariate analysis, please prepared a socio-demographic table (with more factors) for participates by violent category (violent and non-violent). Then compare the means or frequencies between two groups.

• Tables: Please mark results the comparisons by *,**,¶ … for indicate related test and p-value below the tables. Describe the signs below the table.

• Table 1: P-values conducted from which statistical analysis? OR or Chi-squre/Fisher-exact tests? Mention to it below the table (*, †, ¶, etc.).

• Table 1: ORs with zero cell counts can be calculated by correction [2]. Please recalculate the ORs and 95% CI’s for all tables with zero cell counts.

• Table 2: P-values conducted from which statistical analysis? Describe to it below the table by some sign (*, †, ¶, etc.).

Discussion:

• The content of this section is repetitive and lacks useful information to conclude the study. The conclusion is poor.

References:

1. Hosmer Jr DW, Lemeshow S, Sturdivant RX. Applied logistic regression. Vol. 398. John Wiley & Sons; 2013.

2. Pagano M, Gauvreau K. Principles of biostatistics, Brooks. Cole, Belmont[Google Sch. 2000;

Reviewer #2: The authors performed the association between T. gondii IgG seropositivity and violent behavior in inmates. The topic is interesting. The method is not novel and the number of participants is small.

Please imply the limitation of your study in a limitation section, before final conclusion.

6. PLOS authors have the option to publish the peer review history of their article (what does this mean?). If published, this will include your full peer review and any attached files.

Reviewer #1: No

Reviewer #2: No

---

## [Author Response · Author response to Decision Letter 0]

27 Jan 2023

Durango, Dgo. Mexico. January 27, 2023.

Dear Editor,

Please find attached a revised version of our manuscript that has been modified according to the reviewers’ comments. In addition, please find below our response to each of the reviewers’ comments on a point-by-point basis. The revised manuscript meets PLOS ONE’s style requirements. We provided the data set as a supplementary file. 

We appreciate the valuable comments of the reviewers, and we hope the revised manuscript may have more success for publication in the journal PLOS ONE.

Kind regards,

Dr. Cosme Alvarado-Esquivel.

Laboratorio de Investigación Biomédica

Facultad de Medicina y Nutrición

Avenida Universidad S/N.

34000 Durango, Dgo. Mexico.

Tel/Fax.: 0052 618 8 271200

Email: alvaradocosme@yahoo.com

 

RESPONSE TO THE REVIEWERS’ COMMENTS

Journal Requirements:

The manuscript was revised regarding the PLOS ONE’s style requirements. 

2. Please provide additional information regarding the considerations made for the prisoners included in this study. For instance, please discuss whether participants were able to opt out of the study and whether individuals who did not participate receive the same treatment offered to participants.

Information about the option to opt out of the study and treatment offered to inmates who did not participate was added (lines 152-153).

We have no grant numbers in our university.

4. In your Data Availability statement, you have not specified where the minimal data set underlying the results described in your manuscript can be found. PLOS defines a study's minimal data set as the underlying data used to reach the conclusions drawn in the manuscript and any additional data required to replicate the reported study findings in their entirety. All PLOS journals require that the minimal data set be made fully available. 

Upon re-submitting your revised manuscript, please upload your study’s minimal underlying data set as either Supporting Information files or to a stable, public repository and include the relevant URLs, DOIs, or accession numbers within your revised cover letter. 

Important: If there are ethical or legal restrictions to sharing your data publicly, please explain these restrictions in detail. 

Data set was provided as a supplemental file.

No ethical or legal restrictions to sharing the data publicly exist.

Thank you for your valuable comments for improving our manuscript.

Reviewers' comments:

Reviewer #1: 

Abstract:

1. Start the results with some descriptive statistics (mean ± SD, N (%), etc.) for participate (128 inmates).

Information about mean age ± SD and range was added (line 28).

2, Please report more statistical results (coefficients, 95% CI’s, P-value, etc.) in abstract. Statistical results of associations.

More statistical results were reported (lines 38-40).

3. The actual P-value* should be expressed (P=.04) rather than expressing a statement of inequality (P<.05), unless P<.001.

The P value was expressed as P=0.55 (line 40).

4. Methods: Are you measured the costs? Mention to it.

Information about the costs was added (lines 119-120).

5. What mean the “robust association”?

The word “robust” was not used.

Materials & Methods:

6. All acronyms/abbreviations must be explained in parenthesis after their first occurrence within each standalone section of a paper (abstract, main text, acknowledgements, figures, and tables). If any acronyms/abbreviations are used in the main text (excluding tables and figures), please compile them in an "Abbreviations" section at the end of the paper.

A list of abbreviations was added (lines 233-239).

7. Are you test the normality distribution for your quantities responses (HCR-20, Crimes, and AGO score). How you test the normality of data?

Normality was tested for AGQ scores, and information of the test used was added (lines 143-145). 

8. Why did you not use multiple logistic regression analysis to examine the adjusted effects of confounding variables such as age and gender? Reported coefficients are crude effects and the effect of confounders in your study (statistical analyses) is ignored, which causes results with error [1].

Logistic regression analysis was added (lines 140-143 and 177-178).

Results:

9. Report means of quantities variables with standard deviations in the text.

“Mean ± SD”

Means were reported with standard deviations (lines 39, 171-172).

10. For univariate analysis, please prepared a socio-demographic table (with more factors) for participates by violent category (violent and non-violent). Then compare the means or frequencies between two groups.

A new table with sociodemographic characteristics of the study population was added and mentioned in the text (lines 120-121).

11. Tables: Please mark results the comparisons by *,**,¶ … for indicate related test and p-value below the tables. Describe the signs below the table.

Signs to indicate related tests and P values were added to the Tables. 

12. Table 1: P-values conducted from which statistical analysis? OR or Chi-squre/Fisher-exact tests? Mention to it below the table (*, †, ¶, etc.).

Signs to indicate related tests and P values were added to the Tables. 

13. Table 1: ORs with zero cell counts can be calculated by correction [2]. Please recalculate the ORs and 95% CI’s for all tables with zero cell counts.

Odds ratios with zero counts were recalculated by correction. 

14. Table 2: P-values conducted from which statistical analysis? Describe to it below the table by some sign (*, †, ¶, etc.).

Signs to indicate related tests and P values were added to the Tables. 

Discussion:

15. The content of this section is repetitive and lacks useful information to conclude the study. The conclusion is poor.

Repetitive information in the Discussion section was deleted. 

Further discussion of our results was added (lines 214-220). 

The conclusion was modified (lines 228-231).

Thank you for your valuable comments for improving our manuscript.

Reviewer #2: 

1. Please imply the limitation of your study in a limitation section, before final conclusion.

A section regarding the limitations of the study was added (lines 221-225).

Thank you for your valuable comments for improving our manuscript.

---

## [Decision Letter · Decision Letter 1]

27 Feb 2023

PONE-D-22-31301R1Survey on the association between Toxoplasma gondii infection and violent behavior in inmatesPLOS ONE

Dear Dr. Alvarado-Esquivel,

Thank you for submitting your manuscript to PLOS ONE. After careful consideration, we feel that it has merit but does not fully meet PLOS ONE’s publication criteria as it currently stands. Therefore, we invite you to submit a revised version of the manuscript that addresses the points raised during the review process.

We look forward to receiving your revised manuscript.

Kind regards,

Masoud Foroutan, Ph.D; Assistant Professor

Academic Editor

PLOS ONE

Journal Requirements:

Reviewers' comments:

Reviewer's Responses to Questions

**Comments to the Author**

1. If the authors have adequately addressed your comments raised in a previous round of review and you feel that this manuscript is now acceptable for publication, you may indicate that here to bypass the “Comments to the Author” section, enter your conflict of interest statement in the “Confidential to Editor” section, and submit your "Accept" recommendation.

Reviewer #1: All comments have been addressed

Reviewer #2: All comments have been addressed

2. Is the manuscript technically sound, and do the data support the conclusions?

Reviewer #1: Yes

Reviewer #2: Yes

3. Has the statistical analysis been performed appropriately and rigorously? 

Reviewer #1: Yes

Reviewer #2: I Don't Know

4. Have the authors made all data underlying the findings in their manuscript fully available?

Reviewer #1: Yes

Reviewer #2: Yes

5. Is the manuscript presented in an intelligible fashion and written in standard English?

Reviewer #1: Yes

Reviewer #2: Yes

6. Review Comments to the Author

Reviewer #1: The changes in odds ratio values are shown in Table 2. However, the calculations and results of some ORs are not clear. For example, in the first row, the predicted value (OR = 0.5*105.5/6.35*17.5=0.4673) is 0.4637, but the reported value does not match it. Please explain to me how you can calculate this odds ratio.

Reviewer #2: The authors revised the manuscript sufficiently according to the comments. In my opinion, it can be acceptable at the current form.

7. PLOS authors have the option to publish the peer review history of their article (what does this mean?). If published, this will include your full peer review and any attached files.

Reviewer #1: No

Reviewer #2: No

---

## [Author Response · Author response to Decision Letter 1]

13 Mar 2023

RESPONSE TO THE REVIEWERS’ COMMENTS

Journal Requirements:

References were reviewed and it is complete and correct. No retracted papers were found.

Thank you for your valuable comments for improving our manuscript.

Reviewer #1: The changes in odds ratio values are shown in Table 2. However, the calculations and results of some ORs are not clear. For example, in the first row, the predicted value (OR = 0.5*105.5/6.35*17.5=0.4673) is 0.4637, but the reported value does not match it. Please explain to me how you can calculate this odds ratio.

OR with zero values were recalculated and corrected. 

Thank you for your valuable comments for improving our manuscript.

---

## [Decision Letter · Decision Letter 2]

27 Mar 2023

Survey on the association between Toxoplasma gondii infection and violent behavior in inmates

PONE-D-22-31301R2

Dear Dr. Alvarado-Esquivel,

We’re pleased to inform you that your manuscript has been judged scientifically suitable for publication and will be formally accepted for publication once it meets all outstanding technical requirements.

Kind regards,

Masoud Foroutan, Ph.D; Assistant Professor

Academic Editor

PLOS ONE

Additional Editor Comments (optional):

Reviewers' comments:

Reviewer's Responses to Questions

**Comments to the Author**

1. If the authors have adequately addressed your comments raised in a previous round of review and you feel that this manuscript is now acceptable for publication, you may indicate that here to bypass the “Comments to the Author” section, enter your conflict of interest statement in the “Confidential to Editor” section, and submit your "Accept" recommendation.

Reviewer #1: All comments have been addressed

2. Is the manuscript technically sound, and do the data support the conclusions?

Reviewer #1: Yes

3. Has the statistical analysis been performed appropriately and rigorously? 

Reviewer #1: Yes

4. Have the authors made all data underlying the findings in their manuscript fully available?

Reviewer #1: Yes

5. Is the manuscript presented in an intelligible fashion and written in standard English?

Reviewer #1: Yes

6. Review Comments to the Author

Reviewer #1: (No Response)

7. PLOS authors have the option to publish the peer review history of their article (what does this mean?). If published, this will include your full peer review and any attached files.

Reviewer #1: No

---

## [Editor Report · Acceptance letter]

30 Mar 2023

PONE-D-22-31301R2 

Survey on the association between *Toxoplasma gondii* infection and violent behavior in inmates 

Dear Dr. Alvarado-Esquivel:

I'm pleased to inform you that your manuscript has been deemed suitable for publication in PLOS ONE. Congratulations! Your manuscript is now with our production department. 

Kind regards, 

on behalf of

Dr. Masoud Foroutan 

Academic Editor

PLOS ONE